# Discharge of groundwater flow to the Potter Cove on King George Island, Antarctic Peninsula

Ulrike Falk[1,2] and Adrián Silva-Busso[3,4]

[1]Climate Lab, Institute for Geography, Bremen University, Germany
[2]Center for Remote Sensing of Land Surfaces (ZFL), Bonn University, Germany
[3]Faculty of Exact and Natural Sciences, University Buenos Aires, Argentina
[4]University of Buenos Aires (UBA), Buenos Aires, Argentina

*Correspondence to:* Ulrike Falk (ulrike.falk@gmail.com)

**Abstract.** There is only a small number of recent publications discussing glacial runoff in Antarctica and even fewer of them deal with the groundwater flow discharge. This paper focuses on the groundwater flow aspects and is based on a detailed study performed on a small hydrological catchment, informally called "Potter Basin", located on King George Island (KGI; Isla 25 de Mayo), South Shetland Islands, at the northern tip of the Antarctic Peninsula. The basin is representative for the rugged coastline of the Northern Antarctic Peninsula, and is discussed as a case study for possible future evolution of similar basins further to the South. A conceptual hydrogeological model has been defined using vertical electrical soundings (VES), geological and hydrogeological surveying methods, geomorphological interpretation based on satellite imagery, permeability tests, piezometric level measurements, meteorological, geocryological and glaciological data sets. The transmissivities of the fluvial talik aquifer and suprapermafrost aquifer range from 162.0 to 2719.9 $\times 10^{-5}$ m$^2$ s$^{-1}$, and in basaltic fissurated aquifers from 3.47 to 5.79 $\times 10^{-5}$ m$^2$ s$^{-1}$. The transmissivities found in the active layer of hummocky moraines amount to 75.23 $\cdot 10^{-5}$ m$^2$ s$^{-1}$, in sea deposits to 163.0 $\times 10^{-5}$ m$^2$ s$^{-1}$, and in the fluvioglacial deposits they were observed between 902.8 and 2662.0 $\times 10^{-5}$ m$^2$ d$^{-1}$. Finally, the groundwater flow discharge was assessed to 0.47 m$^3$ s$^{-1}$ (during austral summer months January and February), and the total groundwater storage was estimated to 560 $\times 10^3$ m$^3$. The Antarctic Pennsula region has experienced drastic climatological changes within the past five decades. Under the IPCC scenarios a further warming of the polar regions can be expected as polar amplification of our changing climate. Although the basin in consideration is small and results are valid only during austral summer with surface air temperatures above the freezing point, it serves as model study that can be regarded as representative for the western coastline of the Antarctic Peninsula further South under expected future warming with surface air temperatures periodically surpassing the freezing point. This data can be used to adjust glacial mass balance assessments in the region, and to improve the understanding of coastal sea water processes and their effects on the marine biota, as a consequence of the global climate change.

## 1   Introduction

A review of climatological data records over approx. four decades for the Northern Antarctic Peninsula (AP) by King et al. (2017) yield mean annual surface air temperatures of between -2°C and -5°C. Though winter mean values are within -6°C

to -10°C, summer mean values are clearly above melting point with 0°C to +2°C. The AP is an area of high sensitivity to global warming due to its maritime surroundings and its exposure to the low-pressure systems encircling the greater Antarctica (King et al., 2003). Rising temperatures affect the biosphere as well as the evolution of abiotic systems, in particular the dynamics of glaciers (Cook et al., 2005), and the groundwater and surface phases of the water cycle. Climate change has been

characterized by rising temperatures of approximately 2.5°C on the West AP (Ferron et al., 2004; Turner, 2004; Steig et al., 2009). In particular at the end of last century, the average temperature rise has been the greatest in the Eastern region of the AP and specially in the James Ross Archipelago (Skvarca et al., 1998). Barrand et al. (2013) reported an increase in mean annual air temperature of 2.3°C over 4 decades for the Russian Bellingshausen station on Fildes Peninsula, about 10 km away from Carlini Station (formerly called Jubany station). This trend has currently come to a halt, but interannual variability is reported

as high (Falk and Sala, 2015a; Falk et al., 2018a). The western coast of the AP is impacted directly by humid and relatively warm air masses from the Pacific, carried by persistent strong westerly winds (Fernandoy et al., 2018).

    Glaciological changes have been observed along the whole length of the AP's western and eastern coasts. Studies along the AP show extensive glacier acceleration and thinning (De Angelis and Skvarca, 2003), retreat of glacial grounding line and calving front (e. g. Rignot et al. (2011); Rau et al. (2004)). Over the past 5 decades, a general retreat and disintegration of

the AP's cryosphere has been reported by various studies (Falk et al., 2018a; Rott et al., 1996; Skvarca et al., 1998; Scambos et al., 2000; Shepherd et al., 2003; Skvarca et al., 2004; Scambos et al., 2008; Braun et al., 2009). The ongoing change in atmosphere and cryosphere are directly related to extensive changes in the adjacent ocean (Meredith and King, 2005). Polar marine species are highly sensitive to environmental conditions and variability in the AP region (Smith et al., 1999; Peck et al., 2004; Ducklow et al., 2007; Clarke et al., 2007; Montes-Hugo et al., 2009; Schloss et al., 2012; Quartino et al., 2013;

Abele et al., 2017). Glacial melt water input to the coastal systems significantly changes Changes in physical and chemical properties, as salinity, turbidity or light transmission and trace metals are directly linked to glacial melt water input to coastal systems (Braeckman et al., 2021; Henkel et al., 2013; Sherrell et al., 2015).

    At present, the intertidal marine zone in the outlet of the Potter basin (lower than 3 m a.s.l.) is composed of fine sand, silt and a gravel beach ridge with moderate permeability. The sedimentary deposits originated mainly from volcanic clasts,

whereas residues of sedimentary paleozoic and plutonic clasts were observed only in marine deposits. This is an important fact in order to understand the local importance of glacial processes in the hydrology of the Potter basin. The palaeontological and palaeogeographical evidences indicate that penguin rookeries were already present on the cliffs of the Potter Peninsula at the time of marine beach sedimentation (Del Valle et al., 2002). On the basis of radiocarbon dating of organic material found *in situ* it was found that sedimentation of beach deposits occured about 4540 to 4450 years BP. Taking into account reservoir

effects, this was corrected to 1400 years BP. This evidence suggests periods of open seawater in which mammals and marine birds were able to reach and inhabit the coastal areas of Potter Peninsula area in the middle of the Holocene, and that might coincide with a cooling period around Antarctica within the period between 8000 to 4000 years BP, and a *climate optimum* for the AP in the time period of 4000 to 3000 years BP, as suggested by ice core analysis (Del Valle et al., 2002).

    The Potter stream basin is divided in two main channels that have been informally named and will be referred to in this

paper as North Potter (NP) stream and South Potter (SP) stream. Melt water discharge from the Fourcade Glacier supplies for

this hydrogeological system. It is a polythermal glacier forming a part of the Warszawa ice field (Falk et al., 2018b). The two Potter streams have been considered as one basin (Potter Basin) because of the two streams similar geology and to facilitate the hydrogeological analysis. Falk et al. (2018b) assert the importance of the present glacial retreat and the role of the active permafrost as the main drivers of surface and sub-surface hydrological processes in the Potter basin. Our aim is to increase the knowledge and understanding of the consequences of climatic change in the study region in especially the response of the permafrost to a changing hydrologic regime.

During the austral summer mean air temperatures are frequently above zero and snow melting occurs. In some cases, the water intake of the basins comes not only from glaciers but also, to a great extent, from snow melting of the ice-free land areas. On occasions, the snow cover in the non-glacier areas completely disappears and the flow from the active layer and the thawing of the top of the ice-rich permafrost are the main sources of water the feed streams through the saturated zone. One of the characteristics of aquifers in subpolar zones is the specific temporal and spatial pattern of water occurrence. During austral summer, the increase in the depth of the active layer allows more liquid water to be stored in the sediment. The increase in soil temperature results in an increase of melt water runoff from snow, glacier ice and permafrost towards the streams. If the soil temperature reaches higher average values than in previous years during the respective period, the depth to the active layer increases until it can get hydrologically disconnected from the streams and these are left with almost no water flow. There may also be anomalies related to lithological variations giving rise to sectors with varying degrees of vertical and horizontal connection (Yershov, 2004; French, 2017). Hydrogeological studies of permafrost effluent systems in Antarctic environments Silva-Busso (2009) has demonstrated the existence of ephemeral water systems active only during the summer period, derived from spring snow melt and the flow from unconfined aquifers. Three main types of aquifers can be distinguished in subpolar regions depending on their relative position with regards to the permafrost layer: suprapermafrost layers are those above the permafrost, which is their impermeable base; intrapermafrost layers develops within the frozen soil layer; and the infrapermafrost layer develops below the frozen layer, generally at greater depth, where geothermal gradients maintain water in liquid state. Hydrological basins in the study region normally have mixed characteristics. They contain glacial discharge flow, a significant variable flow from rain or snow fall and in some cases flow due to ablation of permafrost. Generally, one of these processes dominates the others. For example, at the start of the austral summer, snow melt is predominant, whereas at the end of the summer, it is the flow contribution from buried ice and the talik zone that becomes prevalent. In between the start and the end of summer all other situations are possible. Fourcade (1960) gives a sketch of a geological map of Potter Peninsula.

This study analyses the direct relation of groundwater discharge and hydrogeological processes to glacier ablation. The aim of this paper is to establish a conceptual hydrogeological model for this processes based mainly on the field work conducted in the austral summer of 2011 in the area of Potter Peninsula.

## 2   Study area

Our study area is part of the Potter Peninsula (62°14'10"S and 58°39'01"W), a non-glaciated area of approximate 6.6 km$^2$ at the SSW tip of King George Island (KGI), the largest of the South Shetland Islands. KGI is located near the North-Western tip

of the Antarctic Peninsula (Fig. 1). It has an elongated shape lying in an approximate NE to SW direction, and about 90% of the island surface is covered with ice. While the island has a steep terrain with a maximum height of 720 m a.s.l. (Rückamp et al., 2011), on the Potter Peninsula area the maximum elevation (210 m a.s.l.) corresponds to Three Brothers Hill.

The Potter Basin is characterized by different geologic units referred to here as cycles, which can be separated into two different groups: volcanic rocks and quaternary sedimentary rocks. The volcanic outcrops are the oldest units and they consti-tute the glacial bedrock. These rocks are theolitic basalt, andesitic volcanic and intrusive rocks. Pankhurst and Smellie (1983) have described five rock samples from the Potter Peninsula with an age between 42 - 48 Ma. The maximum age is provided by the analysis of a cross-cutting andesite plug at Three Brothers Hill, dated by (Watts, 1982) to about $47 \pm 1$ Ma (obtained approximately 2 km away from the Potter Basin). It constitutes a fissured aquifer and it is representative for the area. The second group consists of sedimentary rocks like till deposits (moraines or hummocky moraines), fluvioglacial deposits, lake and marine deposits.

Several authors have analyzed the quaternary deposits in Potter Peninsula (e.g. John and Sugden (1971); Birkenmajer (1998); Del Valle et al. (2002); Wiencke et al. (2008)). Tatur and del Valle (1986), who worked in the paleolimnological and geomor-phological aspects, and Birkenmajer (1998) proposed that the glacial deposits in Potter Peninsula could be of different glacial episodes along the Quaternary. Heredia Barión et al. (2019) give a detailed study on sedimentary architecture and morpho-genetic evolution of a polar bay-mouth gravel-spit system for Potter Cove.

The till deposits have a distribution that includes gravel with abundant silt and clay in the matrix. During the summer, the active layer is mainly developed in these units with their moderate permeability. The fluvioglacial deposits consist of gravel and sands without silts or clays. These units are the fluvial talik, layers with good permeability, through which melt water flows into the Potter Basin. The layers of lake deposits are very thin in the Potter basin and consist of silt and clay, thus they have very low permeability. With regards to the marine deposits, Del Valle et al. (2002) described a 2.7 m thick mid-Holocene sedimentary succession composed of alluvial fan and marine beach deposits on the South-Eastern coast of the Potter Peninsula. Ermolin and Silva Busso (2008) observe a mean annual soil temperature of -2.0°C which is slightly warmer than elsewhere on the AP. Lisker (2004) proposes an average homogeneous geothermal gradient for the AP of 0.03°C considered representative for continental Antarctic areas. Based on this information, the maximum thickness of permafrost in the hummocky moraines and side blackberries is estimated to between 70 - 80 m.

The Potter Basin contain areas of volcanic rocks, moraines of different events, marine and fluvioglacial deposits (see Fig. 2). As noted by Fourcade (1960), features are mainly hummocky moraine with different ice content and the latter is determined as of different stages of genesis (Silva Busso and Yermolin, 2014). Although there are still missing elements to propose suitable chronological order, the oldest units are undoubtedly older basaltic-andesitic or volcanic rock as dated by Pankhurst and Smellie (1983). These are present and intensely fissured in many places, covered by the cryoeluvium product of cryoclastic weathering. They have also been identified in the beds of the river channels in the middle of the Southern Potter Basin (Fig. 1 and 2), which underlines the regional continuity. Fourcade (1960) generally interpretes the till deposits as hummocky moraine and does not differentiate each imposed lithology. The moraine in the South to South-Western part of the map (see Fig. 2) can be assumed to be the lateral boundary of the complete Potter basin. It is the oldest deposit identified here as cycle 1, and probably the

former glacier front or side moraine. The remaining moraines noted on the map are interpreted as hummocky moraines. These moraines have variable ice contents (including buried ice) that originate at least from three different periods of glacial advances and retreats. The hummocky moraines (defined as cycle 1 in, Fig. 2), are called "old till" but they include deposits generated by other more recent processes. They are part of the lower basin above approx. 15 m a. s. l., and have a discontinuous permafrost

with interstitial ice content of about 12-14% (Silva Busso and Yermolin, 2014). This area is, where the active layer and the suprapermafrost aquifer is formed.

While it is difficult to establish a precise limit and also taking into account that the active layer can be developed without a free aquifer French (2017); Silva-Busso (2009), these processes have rarely been observed above 45 m a. s. l. . The more recent hummocky moraines (cycle 2, see Fig. 2) cover a large area from north to south in the middle Potter basin at elevations

between 45 - 60 m a. s. l. . These have continuous permafrost with little development of the active layer (to the north and at lower levels) with a higher content of ice up to 20% and buried ice Silva Busso and Yermolin (2014).

Due to these considerations, it can be assumed, that there is a thin active layer and no development of a suprapermafrost aquifer. Finally, recent deposits called here hummocky moraines (cycle 3 in Fig. 2) can be found from North to South in the upper Potter basin, bordering the Fourcade Glacier at about 60 - 80 m a. s. l.. This permafrost is continuous, with virtually no

development of an active layer. The buried ice abounds no rinterfingering of sediments and ice. The fluvioglacial plains are located between 3 - 15 m a. s. l.. However, their development is almost simultaneous to the genesis of hummocky moraines (cycle 1 deposits, see Fig. 2). The deposits are the fluvioglacial product of rework and outwash of fine material contained in the moraines. This is the geologic unit where the open fluvial talik (French, 2017; Silva-Busso, 2009) is developed with little or absent ice and a non-zero sediment porosity. This layer is identified as the true suprapermafrost aquifer in the study area.

Two sediment units were associated with fluvial rework in the North and South Potter creeks. The marine deposits between 0 - 3 m a. s. l. in the Potter basin are very recent, and no records of marine paleo-deposits were found by Del Valle et al. (2002). These identified with the processes of tidal and wave rework (the latter of lower intensity) and originate from clastic supply from moraines.

The permafrost found here is comparatively warm (mean annual ground temperatures are greater than -2.0°C) and thin (less

than 80 m). Biskaborn et al. (2019) give an overview of global permafrost temperatures and observed trends. They report permafrost temperatures for sites above the Arctic circle and the Antarctic main land of around -10°C, but only about -2°C for the Northern Antarctic Peninsula and a trend of approx. 1°C, per decade. Assuming a homogeneous medium, a geothermal gradient of 0.03°C/m is considered representative for continental Antarctic areas (Lisker, 2004). This yields a maximum permafrost thickness of about 70 - 80 m in bottom moraines and terminal Holocene moraines in the study area, and about

40 and 60 m in the present moraine near the border of Fourcade glacier. In the fluvioglacial plain, gravity slopes (that are not solifluction slopes) and wetland, the permafrost thickness is estimated to less than 40 m. Ermolin and Silva Busso (2008) found the beach zones on KGI to act as limits to the extension of permafrost.

The depth where temporal thawing takes place is determined by the climatic boundary conditions and different lithology types. It is directly linked with the formation of surface and suprapermafrost water. Ermolin and Silva Busso (2008) suggest

that the development of permafrost and groundwater flow is determined by the periglacial geoforms and cryogenic processes.

The collected data show that the thickness of the active layer ranges from 0.5 to 0.8 m in the hummocky moraines. It reaches 1.0 to 1.5 m in the fluvioglacial plain and gravity slopes. On the periphery of the fluvial talik anbd the coastal wetland located at ca. 2 m a.s.l., the influence of surface and groundwater causes an increase of the active layer thickness to 2.5 m.

A prominent feature of the study area are the ice-free areas and above-melting-point summer air temperatures, which leads
to exposed ground surfaces during summer months that are prone to austere cryogenig and exogenic processes, e. g. frost jacking and sorted stripe, typical for the different sediment types of the study area. Frost jacking is a characteristic for slope desposits and wetlands, and it is prevailing in the study area's bottom moraines of presence and Holocene that contain the suprapermafrost aquifer. The characteristic feature in the fluvioglacial plains, wetlands and fine-textured bottom moraines, is the sorted stripe visible as nets with a pattern size of about 0.5 to 1.0 m. Short-term diurnal turnover in soil temperature around
freezing point temperature in the wetlands and fine clastic soils found mostly in lake depressions result in a smaller-size pattern. Slope deposits and lateral moraines that contain ice-rich permafrost and buried ice stem primarily from local bedrock subjected to landslides. The intense weathering and highly dynamic processes of freezing and thawing in addition to the observed high wind speeds in the study area, lead to formation of colluvial slopes and vertical cliffs. Ermolin and Silva Busso (2008) identify thawing of ice-rich permafrost and buried ice as the agents to the prevailing processes of thermoerosion in fluvioglacial plains
as well as lake formations. In general, the fluvioglacial deposit areas present slightly lower gradients, from which it is possible to infer a higher permeability. These units have gravel sediments with few fine sediment content. The volcanic rock outcrop areas are fissured aquifers and show significantly lower gradients, from which it is possible to infer a lower permeability. These areas are distributed throughout the subsurface region including Potter basin.

Silva-Busso (2009) found that the groundwater discharge from flow from suprapermafrost and interpermafrost aquifers to be
the main drivers of the hydrodynamic balance in the Potte catchment. They also found, that streams and lagoons in a close-by small hydrological basin (Matías Basin) are supplied by the suprapermafrost aquifers.

## 3   Data and methods

Several analytic methodologies have been applied including *in-situ* observations and satellite remote sensing data in order to establish a comprehensive overview on groundwater discharge, hydrological processes and their relation to glacier ablation.
A Quick Bird image was used for correcting the geologic and geomorphologic interpretation of the Potter basin from field data. Figure 1 shows the basin boundaries and divides, as well as watercourses, derived from a topographic map of the area (Lusky et al., 2001) and complemented by aerial images and own *in-situ* GPS data. The instrumentation used for this study included a resistivity meter Fluke 80 (Fluke Corporation, Netherlands) to measure the electrical current and potential difference to calculate electrical resistivity which allows for a quantification of water and/or ice content distribution in the respective layer.
Conductivity and pH-values were obtained with a conductivity meter WTW315 and a pH meter WTW320 (Xylem Analytics, New York, USA). The water depth was measured with a piezometric sonde. Optical levelling was conducted with a FOIF 57 (FOIF Co, Suzhou, China), and distances were measured with a Telemeter Nikon Forestry Laser PRO 550 AS (Nikon Inc., Japan). All observational and measurement sites are mapped in Fig. 1.

Geologic units were mapped and the hydrogeology aspects were analyzed using vertical electrical soundings (VES) as local geologic and hydrogeologic survey methods. The VES were conducted using a digital resistivitymeter with manual telluric compensation. The sensitivity of the device is 0.01 mV and 0.1 mA, using a continuous power supply of 160 V and 1000 mA. A Schlumberger geoelectric line device with an electrode spacing of $AB/2$ = 44 m was used, with $AB$ being the maximum electrode spacing (in all VES) to investigate the local quaternary deposits. The electric resistivity is an indirect measure of changes in water, ice and rock. The results were correlated with the ground cuts in the stream crankcases and wells that were drilled for the recognition of the suprapermafrost aquifer. The subsurface geology can be assessed by geoelectrical studies. The geoelectric techniques used are indirect methods to identify changes in ice content, water salinity or lithological change and have been previously applied and validated for subsurface studies in Antarctica Muñoz Martín et al. (2000); Silva Busso et al. (2013). Eighteen VES were performed in the Potter basin (called VES12 - VES29, see Fig. 1) and interpreted. From these profiles, five resistive horizons and eight resistive layers were identified. Tables 1 and 2 summarize the characteristics of the found resistive layers, i.e. layer thickness and resistivity, and the geological interpretation. This method provides the layers thicknesses, depth, and distribution of underground ice or water content. It is the basis for the estimation of the hydrogeological cross sections of the outlet and the mid basin. The location of VES surveys, permeability tests and static level groundwater measurements points in Potter basin are shown in Fig. 1. From the interpretation of these data (see Tab. 1), a hydrogeological model was derived (see section 2).

Thirteen wells were drilled with a hand drilling system for the deep groundwater level measurements and the slug test, which measures the permeability by the salinity method (Custodio and Llamas, 1983). The wells were between 1.25 to 2 m deep, and can be divided into two major groups: coastal wells that are drilled on terrain lower than 3 m a. s. l. and wells on higher inland terrain. Soil samples were taken and analysed from 10 wells. The first group shows thick dark grey sands composed lithologically of fragments of volcanic rock in the layer above 1.25 m, and fine gravel, dark grey sands and basaltic lithic composition below. The second group shows fine gravel, dark grey sands and basaltic lithic composition between 0 to 1.75 m. The lithology of both groups is very uniform and is almost entirely basaltic volcanic composition. Observations of specific permeability by slug tests were conducted at the locations shown in Fig. 1 (see permeability test points) identifying layers with low and high permeability (i. e. fluvioglacial deposits) according to:

$$K = V \cdot \phi \cdot \nabla_i, \tag{1}$$

where $K$ is the permeability, $V$ the Darcy velocity, $\phi$ the specific yield and $\nabla_i$ the hydraulic gradient.

The fluvial talik is the layer showing the highest permeability. Since the water in the active layer is fresh water (ca. 400 $\mu$S cm$^{-1}$), another test to quantify the permeability was conducted with NaCl-tracer NaCl (Custodio and Llamas, 1983; Hall, 1996). The point permeability test yielded a porosity value of 14 - 20%. Results are shown in Fig. 3. Total and specific porosity were obtained following the method proposed by Schoeller (1962) and Custodio and Llamas (1983). The field samples of the classic deposits were analysed in the laboratory using granulometric technique. Results are listed in Table 3. The samples of

the cryoeluvium gave dispersed results above 0.4. The value of specific yield or porosity for fissured basalt was estimated from the difference in weight between the dry and water-saturated sample.

In less permeable layers (like moraine deposits and basaltic rocks), the slug test was performed following the Lefranc method with variable levels (Hall, 1996):

$$K = \frac{d^2 \ln(2L/d)}{8Lt} \ln \frac{h_1}{h_2}, \tag{2}$$

with $h_1$ and $h_2$ the water levels at the start and the end of the test, respectively; $t$ is the time passed for the water level going from $h_1$ to $h_2$; $L$ is the longitude and $d$ the diameter of the drill.

It should be noted that the application of these units, the non-uniformity and poor accessibility of terrain makes it very challenging to construct the wells or conduct a test by traditional pumping.

The data enabled us to derive a suprapermafrost piezometric map of the talik and hydrological active layer area. From the slug test data, the permeability was estimated in the different geologic unit outcrops. This method has been demonstrated to be effective in volcanic rocks and moraine deposits and it is recommended for use on low permeability aquifers. Due to the high permeability found at this research site, this method was not applicable for the fluvioglacial deposits, but instead a salt permeability test was used (Custodio and Llamas, 1983). The groundwater hydraulic gradient was calculated on the basis of the piezometric map of the suprapermafrost aquifer swhich contains different hydrogeological units. The simplest way to to calculate the hydraulic difference between two elevation isolines is to calculate the linear perpendicular distance separating them. The meteorological, permafrost and glaciological data sets were used for a complementary analysis of the hydrogeological model presented in here.

The calculation of the groundwater discharge requires the use of the Darcy's Law, assuming that summer groundwater discharge has completely saturated the sediments and that the runoff from the (adjacent) Fourcade Glacier is high. These assumptions were valid during 1 to 1.5 months in the austral summer (here, January and February) while daytime air temperatures are above melting point for the majority of days per month.

For a proper assessment of underground discharge flow and local hydrodynamics, it is essential to define the hydrogeological units, which depend on the geological deposits present in the basin. To achieve this goal, a geological interpretation was performed on a Quick Bird satellite image (2006), in combination with geological field data taken over multiple campaigns (in austral summers of 2003/04, 2007/08 and 2010/11). However, it is important to define the boundaries of volcanic and clastic geological units in the Potter Basin. To accomplish this, photo-feature limits of the different units were identified corresponding to the satellite image (Satellite Image Projection UTM, Zone 21 South, pixel resolution: 0.6 m; date: 2006.02.16).

## 4 Results

The Potter basin consists of two main channels, North Potter stream and South Potter stream. It is a hydrological system that is mainly driven by the discharge from polythermal glaciers that form a part of the Warszawa Ice Field. In both sub-basins, the

appropriate places for discharge measurements were chosen (see Fig. 1). The contemporary glacial retreat and active permafrost processes are found to be the main drivers of runoff and groundwater processes in the study area as discussed in the following sections.

## 4.1 Hydrogeologic units

The shoreline is formed by a gravel coastal cordon that controls an inland lagoon with deposits of fine sand and silt (Heredia Barión et al., 2019). This is a transition zone between the talik and the cryopeg (Silva-Busso, 2009), although the latter is likely to be replaced by a wedge of fresh water coming from the creeks due to the magnitude of groundwater flow discharge.

Resistive variations are found to be mainly due to lithological changes or changes in ice content. These two factors determine the probability of development of an active layer or a suprapermafrost aquifer. Moreover, cross sections of subsoil can be derived from these data and groundwater discharge sections quantified. This method allows to detect a layer of non-outcropping sediments in the area of the lower Potter basin (Layer V). These sediments are thicker than the other geologic units, e.g. VES 12 profiles show a layer thickness of non-outcropping sediments of 13 m at a base depth of 22 m. These resistive layers can be interpreted as old till deposits of more ancient hummocky moraines or previous fluvioglacial events.

Permeable deposits are interpreted here as fluvial talik that correspond to layers I, III and V, summarized here as Group A (see Tab. 2). Layers III and V showed similar resistivity and can be assumed to be lithologically the same. Layer I has a lower resistivity because it contains marine deposits or salt water due to its close vicinity to Potter cove, however, it is a thawed and permeable zone. Layer II (cryoeluvium) contains deposits and is located on the Southwestern slope of the old moraine. Although water can move by snow infiltration, it is not in contact with the Fourcade Glacier, which is outside of this group despite their similar resistivity. Group B is defined only as the cryoeluvium. Another group (Group C) is defined by the hummocky moraines or discontinuous deposits with low ice content for the layer IV. From the layers of this group evolves the active layer, containing the summer suprapermafrost aquifer outside the talik and the coastal zone. The observed resistivity is significantly higher than for Group A (up to 3689 Wm in VES 28) indicating areas with low ice content and distinguishing it from other areas. Group D is represented by layer VI, attributed to deposits with hummocky moraines with buried ice where an active layer develops discontinuously or not at all, thus it is assumed as a layer of low permeability. Group E contains layer VII which consists of weathered basalt, where an aquifer with low ice content forms. This originates in the polythermal bedrock glacier (Fourcade Glacier) which receives the subglacial water from the whole catchment area. The increased resistivity of igneous rock is evident in this layer. Finally, unaltered basalt is found in layer VIII, with the highest resistivity signal, defining the regional aquifuge (Group F).

Another important parameter to be measured is the permeability of each group with aquifer characteristics (talik and fissurated suprapermafrost aquifer). *In-situ* observations of specific permeability by slug tests and NaCl-tracer were used to identify layers with low and high permeability. The fluvial talik is the layer showing the highest permeability. This is considered to be representative for all layers in Group A. Group A contains layers of psamitic marine deposits whose permeability is less than that of the fluvioglacial deposits (gravels).

Figure 3 presents results of the permeability tests. Table 2 summarizes the interpretation of these results as the local hydrostratigraphic groups and their corresponding permeabilities. In groups without aquifer development no action was performed. Based on the observed geological profiles in the ravine channel and in geological surface surveys, geoelectrical data and permeability tests according to the hydrostratigraphy scheme of the area in the Potter Basin is proposed. The ice concentration in soil samples and the *in-situ* identification of permafrost observed in cross-sections and wells allow to infer the characteristics of the permafrost. Based on the information at surface and subsurface, geological cross sections for the two Potter basin were performed.

Figure 4 shows the longitudinal cross section through the whole basin with its axis from the coast to the glacier front (cross section $AA$' as defined in Fig. 4). Figure 5 displays the cross section downstream through the permeable deposits on the coast ($BB$' section as defined in Fig. 4). These cross sections give an idea of the relationship between the hydrostratigraphic sets, but Fig. 5 is particularly important since it defines the aquifer section of each layer in the discharge area at the coast. Considering the average thickness and permeability, the transmissivity ($T$) of each layer can be calculated (Silva-Busso, 2009). For the talik area, only the areas of fluvioglacial and marine deposits are considered, assuming suprapermafrost sections in the moraines and fissured basalts occur along the entire waterfront. The results are presented in Table 4 and the obtained values are within the typical ranges for such types of lithologies (e. g. Custodio and Llamas (1983)).

## 4.2   Suprapermafrost aquifer

In the lower and medium Potter basin 11 manual drillings of boreholes (see Fig. 1) were made with the purpose of verifying the continuity of the suprapermafrost aquifer. Water depth was measured to produce a piezometric map that allows for the calculation of hydraulic gradients. In the upper Potter basin, manual drilling proved to be impossible due to the existence of continuous permafrost with a strong ice layer and poor development of the active layer. These findings suggest that there is no development of a suprapermafrost aquifer. Considering the geological field observations and analysis of satellite imagery, there are strong indications that above 45 m.a.s.l the possibility of a development of suprapermafrost aquifer is negligible. Therefore, the only detailed piezometric map covers the area of elevation between 0 - 45 m.a.s.l (shown in Fig. 6). The groundwater flow network shown here is well integrated in NW direction following the topographic gradient and discharging into Potter Cove. This piezometric surface is well integrated in Groups A and C (fluvioglacial, marine and hummocky moraine deposits summarized in cycle I, Fig. 2 and Tab. 1). Group E contains the layers of fissured aquifer basaltic rock where no observation wells were drilled (only two for the slug test) making it impossible to identify its extent or its piezometry. There is, however, sufficient hydrogeological field evidence of its occurrence, extent and capacity to store and supply water. Fissured basalt extends into the subsurface throughout the catchment area and receives directly subglacial water discharge. According to Rückamp et al. (2011), ice thickness is approximately 80 meters in the area Warszawa Icefield. The topography gradient of the bedrock between ice-free areas and rock outcrops was determined by geoelectrical observation to range within $5 \cdot 10^{-2}$ - $8 \cdot 10^{-2}$ m/m. The obtained value was used as a first approximation to the subglacial topographic gradient of the glacial bedrock topography. There is a high similarity between measured values of piezometric and topographic gradients in the Potter

Basin and in the adjacent Matías Basin on Potter Peninsula (Silva-Busso, 2009). Here, topographical gradients instead of the hydraulic gradients are used as input.

## 4.3 Relation between groundwater and permafrost

In the Potter basin the permafrost is continuous in the upper basin, discontinuous in the middle basin and in the fluvioglacial deposit areas, while the cryopeg is found in coastal areas. Basal moraine deposits and fluvioglacial deposits are superimposed on volcanic rocks and marine deposits in the coast. On the basis of the VES observations, the thickness of the active layer is calculated to 1.74 m and the depth of the talik to 5.2 m. It is important to validate these values by other methods, since these are crucial values that determine the saturated thickness of groundwater flowing to the cove. One possibility is to use the standard depth of seasonal thawing ($H_d$) using the empirical functions proposed by Khrustalev (2005):

$$H_d = \sqrt{(2\lambda_d\tau_1 t_1/q)^2 - k_{soil}/2q} \tag{3}$$

$$k_{soil} = (0.25 - \tau_1/3600) \cdot (t_1 - t_c) \cdot \sqrt{\lambda_c C_c \tau_1} \tag{4}$$

$$q = \rho \cdot (w_t - w_n) \cdot \gamma_d + (\tau_1/7500 - 0.1) \cdot (c_d t_1 - c_c t_0) \tag{5}$$

$$t_1 = 1.4 t_p + 2.4 \tag{6}$$

$$\tau_1 = 1.15 \tau_p + 360 \tag{7}$$

where $H_d$: the normative depth of seasonal thawing in m; $k_{soil}$: a parameter depending on the climate and soil thermal properties in Kcal hours$^{-1}$ m$^{-2}$); $\lambda_c$: the thermal conductivity coefficient of permafrost in kcal m$^{-1}$ °C$^{-1}$; $\lambda_d$: the thermal conductivity of soil thawed coefficient in kcal m$^{-1}$ °C$^{-1}$; $t_0$: mean temperature of permafrost in °C; $t_p$: positive mean air temperature in °C; $t_c$: frozen temperature in °C; $\tau_p$: duration of the period with positive air temperatures in hours; $w_t$: weight of wet soil in tons; $w_n$: weight of dry soil in tons; $\rho$: latent heat of fusion of ice in kcal/t; $c_c$: heat capacity per volume of frozen soil in kcal m$^{-3}$ °C$^{-1}$; $c_d$: heat capacity per volume of soil thawed in kcal m$^{-3}$ °C$^{-1}$; $\tau_1$: a coefficient normative temperature and $t_1$: normative temperature in °C. This value is defined as the greatest depth observed where the active layer of seasonal thawing stage coincides with the top of the permafrost over the past 10 years. The empirical functions by Khrustalev (2005) were calculated in its original units (Tab. 5) but all other results were translated to SI units.

The normative thawing depth $H_d$ was calculated from the 42-years' meteorological time series at the Russian Bellingshausen Base (Martianov and Rakusa-Suszczewski, 1989; AARI, 2016). From this data time series $t_p$ and $\tau_p$ were obtained. In the period between March and November, which is the austral winter time, the amount of positive degree days (PDD) per month does not always reach 2%. The application of the method after Khrustalev (2005) requires a percentage of PDD per month higher than that value. Falk and Sala (2015a) give an analysis of PDD and melt days in their Fig. 4, showing that occurrence of melt periods during winter months is not unusual. The PDD condition is met during summer time but not always in winter months. The values for the physical properties of permafrost and soils ( $\lambda_c$, $\lambda_d$, $t_0$, $t_c$, $\rho$, $C_c$, $C_d$, $\gamma_c$ and $\gamma_d$) for the study sites were taken from field observations and Khrustalev (2005). The weights of wet and dry soil ($w_t$ and $w_n$) were determined

specifically for the sample wells for the piezometers (see Fig. 1). The results are listed in Tab. 5. In summary, the maximum summer normative thawing depths calculated according to Khrustalev (2005), amount to values between 3.21 m and 4.29 m. Analysis of the geoelectric measurements leads to an estimate of the thawing depth of about 1.74 m to 5.2 m for austral summer months. The two methods show comparable results, meaning that the estimated thickness and the aquifer cross section are plausible. It also shows that, although the PDD-condition in the Khrustalev-method are not completely met, the application of this method yields realistic results.

## 4.4 Hydrogeologic conceptual model and discharge estimation of groundwater flow

In this section, a hydrogeological model is proposed for the groundwater flow discharge from the Potter basin into the cove. The groundwater in the talik zone nearly completely saturates the coarse sediments over which channels are formed. The groundwater static levels in the talik have a depth between 0.25 m and 0.65 m. Thus, the runoff is carried in channels on an almost completely saturated talik zone. The Potter creeks can be regarded as a direct product of glacial discharge with very little supply from the talik into the channel. In addition, rainfall increases the volume of water flowing through these channels significantly. These conditions are met during the period of more or less two months in austral summer and autumn with varying duration (Falk and Sala, 2015a). The hydrogeological observations in this paper refer to the period between 27 January 2011 to 05 March 2011. The groundwater flow circulates in the suprapermafrost aquifer, aquifer talik and fissurated basaltic rock. It is assumed to be a free aquifer for its hydrogeological and geocryological characteristics. A pumping test in the adjacent Matías Basin carried out by Silva-Busso (2009) proves this basin to be a free aquifer, as is inferred for the Potter Basin. First, both aquifers are considered free or unconfined and the specific porosity or specific yield ($\phi_e$) can be calculated by the specific retention ($m_r$) according to Custodio and Llamas (1983):

$$m_r = 0.03(\%\text{sand}) + 0.35(\%\text{silt}) + 1.65(\%\text{clay}) \tag{8}$$

and subtracting it the total porosity ($\phi_t$)

$$\phi_e = \phi_t - m_r . \tag{9}$$

This was done by particle size analysis of samples obtained from the wells (permeability test sites see Fig. 1). The total porosity amounts to $\phi_t = 0.17m^3/m^3$ for Group A and $\phi_t = 0.14m^3/m^3$ for Group C, espectively. The calculated effective porosity is then calculated as $\phi_e = 0.12m^3/m^3$ for Group A and $\phi_e = 0.06m^3/m^3$ for Group C. As for fissured aquifer (Group E) in the lower basin, this may have some degree of confinement since low pressure hydraulic exsurgence (due to the topography) occurs. Thus, free aquifers exist in the fluvial talik of the Potter basin (Group A) that have moderate to high transmissivity (140 to 2350 $m^2/day$) in an aquifer depth of 5.2 m. The specific yield ($\phi_e$) amounts to $\phi_e = 0.12m^3/m^3$ and is limited to the cryopeg and the talik area. The unconfined suprapermafrost aquifer (Group C) has moderate to low transmissivity (65 $m^2/day$), low aquifer depth (1.74 m), and a specific yield of $\phi_e = 0.06m^3/m^3$. Group C has a limited extension to the

middle and lower basin. The fissured aquifer (Group E) is the largest layer that covers the entire catchment area. It has, however, a very small depth of 3 m, and a low transmissivity of 4 $m^2/day$. The hydraulic type is not exactly known. All aquifers form a well integrated piezometric surface with a flow discharge into Potter Cove with hydraulic gradients in the range of 0.029-0.093 m/m. The transversal section of Fig. 4 allows for the calculation of the area of each aquifer sections. With this information, the monthly discharge can simply be estimated by Darcy's law:

$$Q = K \cdot S \cdot \nabla_i, \tag{10}$$

where $Q$ is the flow through the permeable porous section in m$^3$ s$^{-1}$, $S$ is the saturated area in m$^2$, and $\nabla_i$ the hydraulic gradient of the aquifer section in m/m. It is also possible to calculate the Darcy velocity ($V_d$), which can be estimated by an approximation of groundwater flow velocity by

$$V_d = K \cdot \nabla_i, \tag{11}$$

Applying Eq. 10 and 11 for each aquifer results in the average monthly flow in m$^3$ s$^{-1}$ and is listed in Table 6. Based on the analysis for the study period, the set of aquifers (Groups A, C and E) have a monthly average discharge of $Q = 0.47$ m$^3$ s$^{-1}$ into Potter Cove during the austral summer months 2011.

It is noteworthy to analyze the Darcy velocity, as it represents approximate transit times. The total time it takes for a water parcel to traverse the whole groundwater reservoir along the direction of the discharge was measured for each hydrostratigraphic set. In the case of Group A aquifers, the transit time was calculated to up to 14 days, meaning a high velocity of the water flow. For group C aquifers, the calculated velocities are significantly lower with estimated transit times of the order of months to a maximum of 4 months. For a short summer, the observed transit times might not be sufficient for the water flow to cross the aquifer resulting in transit times of over a year. Finally, the aquifers of group E show the lowest velocities and a determined annual flow rate that can be on the scale of decades. In this group, it is important to consider its location at the top of the bed rock of Fourcade glacier.

According to Paterson (1994), polythermal glacier water circulation at the base occurs throughout the year. This means for the fissurated basaltic rock aquifer of group E, that it is the only one with the possibility to recharge and discharge throughout the year. Otherwise, if it is only active during summer, the transit of water through this layer would have significantly longer resident times.

In summary, it is possible to differentiate between three main flow regimes of distinct discharge time scales (daily, monthly and decadal). Finally, the total groundwater storage ($R_t$) is determined by the total water capacity that drains the aquifer gravitationally (Custodio and Llamas, 1983). The storage capacities can be calculated by

$$R_t = A \cdot b_{sat} \cdot \phi_e \tag{12}$$

for each group, where $A$ is the extension area, $b_{sat}$ the saturated thickness, and $\phi_e$ the effective yield.

This allows for estimation of the respective storage terms derived from the effective porosity. The results are listed in Tab. 7. The sum is the total groundwater storage and amounts to $560 \times 10^3$ m$^3$.

## 4.5 Sensitivity analysis of groundwater discharge and storage terms

The characteristics of the applied methods and results suggest the application of a sensitivity analysis of the parameters used to estimate groundwater discharge ($Q$) and total storage terms ($R_t$). The task is to assess the importance of flow in basaltic rock compared to the other flows, and to check the sensitivity of the achieved results depending on the input variability. A *ceteris paribus* approach analyses the impact of one parameter on a reference outcome, holding constant all other parameters. However, in this case it is very difficult to obtain alternative values for comparison. For this reason, it was decided to establish a reference condition, although no longer arbitrary, that allows us to study the possible dispersion of the results. The sensitivity ratio ($S_s$) is given by

$$S_s = (Q_{max} - Q_{min})/Q > 0.5 \tag{13}$$

and

$$S_s = (R_{t,max} - R_{t,min})/R_t > 0.5 \,, \tag{14}$$

where $Q_{max}$ and $Q_{min}$ are the maximum and minimum plausible discharges, and are taken from as the minimum and maximum hydraulic gradient to calculate the minimum and maximum flow values. $R_{t,min}$ and $R_{t,max}$ are the maximum and minimum plausible total groundwater storage terms, respectively.

The most sensitive parameters have been highlighted in Tab. 6 and 7. In the assessment of groundwater flow discharge three sensitive parameters have been detected. The first is the hydraulic gradient of group A, particularly in the fluvioglacial deposits where variations between 1 - 5 $\times 10^{-2}$ m m$^{-1}$ are sufficient to overcome the condition in Eq. 13. However, since these layers exist in a well-defined integrated piezometry with defined hydraulic gradient, the variability of hydraulic gradients are limited and it is therefore considered a robust parameter. The quantified discharge of Group A is also sensitive to the permeability. Values for the permeability vary on a daily basis between 2.31 and 11.57 $\times 10^{-3}$ m s$^{-1}$ in the old till deposit layers. This is a complex issue as this layer is only detectable in deep VES that have not been applied in the field. Therefore, the measured values in the actual fluvioglacial deposits cannot be considered a robust parameter. The permeability could be higher, resulting in a possible underestimation of groundwater storage terms.

The third parameter is the aquifer cross section of the old till deposits. Variations to double or half size of the aquifer cross section fulfill the condition in Eq. 13 for the same reasons mentioned above. It is, however, assumed a robust parameter since the VES data on the basis of which the layer geometry was determined, and the calculation of the depth of seasonal normative thawing were well defined for the considered areas and layer depths. The calculation of total groundwater storage has proven to be sensitive to the parameterization of specific porosity within group E (fissured aquifers). The problem is

related to the type of porosity (fissures) and the difficulty to assess the layer depth. Assuming a higher confinement or higher values of specific porosity as condition in Eq. 13 is rapidly reached. It is difficult to estimate an appropriate value for this parameter. The permeability measured in areas of outcrops give us a similar magnitude, but several authors have mentioned a nonlinear relationship between permeability and porosity (Kozeny, 1927; Carman, 1938). A value of $\phi_e = 0.01$ m$^3$ m$^{-3}$ has been applied according to the empirical relationship proposed by Bourbié et al. (1987). The lithologies described there do not match completely with the ones found in our study area, due to the contact with basaltic rock of the bedrock that can be altered and possess a certain degree of porosity. This discontinuity is typical of basaltic rock. Despite these inconsistencies, the approach by Bourbié et al. (1987) is seen as the best reference. This parameter can thus not be considered as very robust. In all cases, the condition stated in Eq. 14 is fulfilled.

## 5   Discussion and Conclusions

There are several conclusions from this work related to the groundwater discharge in Potter Cove (geology, permafrost conditions, glacial water supply and others). The applied methodology is simple and represents a classical groundwater study in the field, as well as applied criteria and concepts that are well proven in hydrogeology. First, the Potter basin is a very young basin and the volcanic rocks contain substrate with a very low permeability. Above these are glacial or fluvioglacial clastic sequences. The latter correspond to different progression and retreat events of the Fourcade Glacier in recent times. As a result of these processes, the discontinuous permafrost occurs only in the lower and middle basin and continues at the top of the upper basin. The discontinuous permafrost zone, talik fluvial areas, cryopeg and active layer constitute the local aquifer suprapermafrost. Based on the geological profiles observed in the ravine channel and in hydrogeological surveys, we propose a hydrogeological schematic of the area in the Potter Basin on King George Island shown in Fig. 7.

It has been possible to define a separate hydrostratigraphy in clastic (Groups A and C) and fissured (Group E) aquifer layers. All of them allow for groundwater movement towards Potter Cove, although with different flow velocities, between 0.08 to 41.3 $\times 10^{-5}$ m s$^{-1}$. Transmissivities were estimated as 75 to 2720 $\times 10^{-5}$ m$^2$ s$^{-1}$ for clastic aquifers and 4.63 $\times 10^{-5}$ m$^2$ s$^{-1}$ for fissured aquifers. Based on their geology, these aquifers can be considered as unconfined aquifers. The groundwater discharge in Potter Cove is estimated to be about 0.47 m$^3$ s$^{-1}$, and the total groundwater storage to be about 56 $\times 10^4$ m$^3$. The conceptual model shows a sensitivity to the parameters of permeability, hydraulic gradient and aquifer section for Group A (i.e. old till deposit). This layer has only been investigated by geoelectric methods, which increase the uncertainty of the values. For group E (i.e. fissured basalt), the model also shows a higher sensitivity to variations in specific porosity. This is a very variable parameter that is difficult to obtain in fissured aquifers. The hydrogeological model helps to understand the groundwater discharge into Potter Cove with a plausible interpretation of the local hydrostratigraphics aspects. It also facilitates the understanding of other parameters such as changes in salinity and turbidity in relation to discharges in Potter Cove. The application of Darcy's law is valid in the talik and active layer zones (i. e. suprapermafrost aquifers) considering that all sediments are saturated with water during austral summer, here January and February (including the period of our field data 27 January 2011 to 05 March 2011).

The groundwater flow values obtained here are thus considered as a significant contribution to the local groundwater discharge into Potter Cove. The waters from the aquifers of the Potter creek basin quickly reach the coast of Potter Cove along with the glacial melt water runoff, and greatly modify the properties of coastal sea waters. The supply of fresh water to the marine zone has an impact on the biota mainly due to changes in salinity, temperature and sediment loads affecting the biota's local environment. The groundwater discharge and melt water runoff is a consequence of the ablation of glacial ice and the retreat of the inland ice caps as a result of ongoing climate change processes. For this reason, the estimation of the groundwater discharge during the summer further the understanding of changes in the biological changes in Potter Cove related to the freshwater supply. This work gives an estimate and quantification of the summer groundwater flow discharge that finally reaches Potter Cove, and a description of a typical geohydrological system for the Northern West AP. The coastal zones are important for biological diversity, and are very sensitive to observed recent climate change in maritime Antarctica. Glacial meltwater discharge is regarded as a key parameter that impacts marine environment, ocean circulation and benthic and pelagic ecosystems (Meredith et al., 2018; Braeckman et al., 2021). The year 2016 was year with long glacial melt periods observed during austral winter. The year 2016 showed an extreme a strong El Niño event. For the AP it can be regarded as representative for future scenarios under ongoing climate change. Assuming a further warming along the Antarctic Peninsula, Potter Cove in the South Shetland Islands can be regarded as a representative case study on future scenarios for the coves along the AP southwards.

## 6   Code availability

All R codes and calculations are available on request from Ulrike Falk and Adrián Silva-Busso.

## 7   Data availability

Supplementary data are available by Falk et al. (2015); Falk and Sala (2015b); Falk et al. (2017) and at: https://doi.org/10.1594/PANGAEA.8 http://dx.doi.org/10.1594/PANGAEA.848704.

*Author contributions.*  UF was the PI of the glaciological and climatological work package within the IMCOAST project, and also leading the glaciological work on KGI within the IMCONet project. ASB was the PI of the hydrological work package within IMCOAST. All final analysis and post-processing of hydrological and geological data sets was performed by ASB. The manuscript was written by UF and ASB.

*Competing interests.*  There are no potential conflicts of interest regarding financial, political or other matters.

*Acknowledgements.*  This work has been possible thanks to the support of the following institutions: Instituto Antártico Argentino de la Dirección Nacional del Antártico (IAA-DNA) in Argentina, the Center for Remote Sensing of Land Surfaces (ZFL) at the University of Bonn in Germany, Instituto Nacional del Agua (INA) and Buenos Aires University (UBA) in Argentina. The assistance given by the overwintering

staff from the Ejército Argentino at Carlini Station and the logistic assistance provided by the Alfred Wegener Institute (AWI) is very much appreciated. This paper is the result of the cooperative research within the framework of the IMCOAST project (BMBF award AZ 03F0617B) and the IMCONet Project (FP7-PEOPLE-2012-IRSES).

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

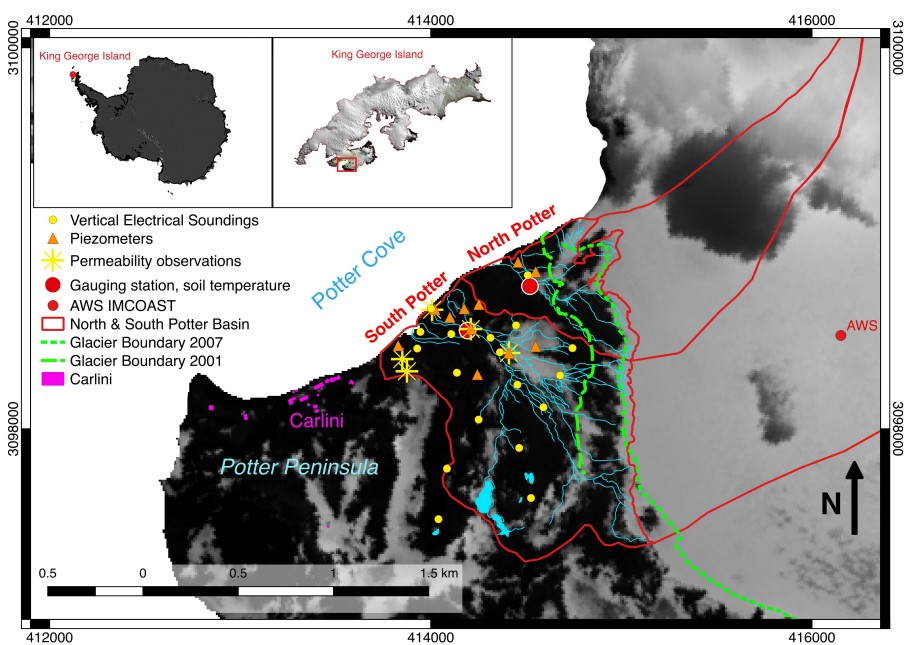

**Figure 1.** Map of research area on the Potter Peninsula, King George Island (Northern Antarctic Peninsula). Shown are the locations of the installations of vertical electrical soundings, piezometers, observations of permeability, gauging stations and soil temperature measurements and the automated weather station (AWS, Falk and Sala (2015a)). The hydrological basin boundaries of the North and South Potter creeks with their drainage channels are also displayed. Background: SPOT-4, 18 November 2010, ©ESA TPM, 2010

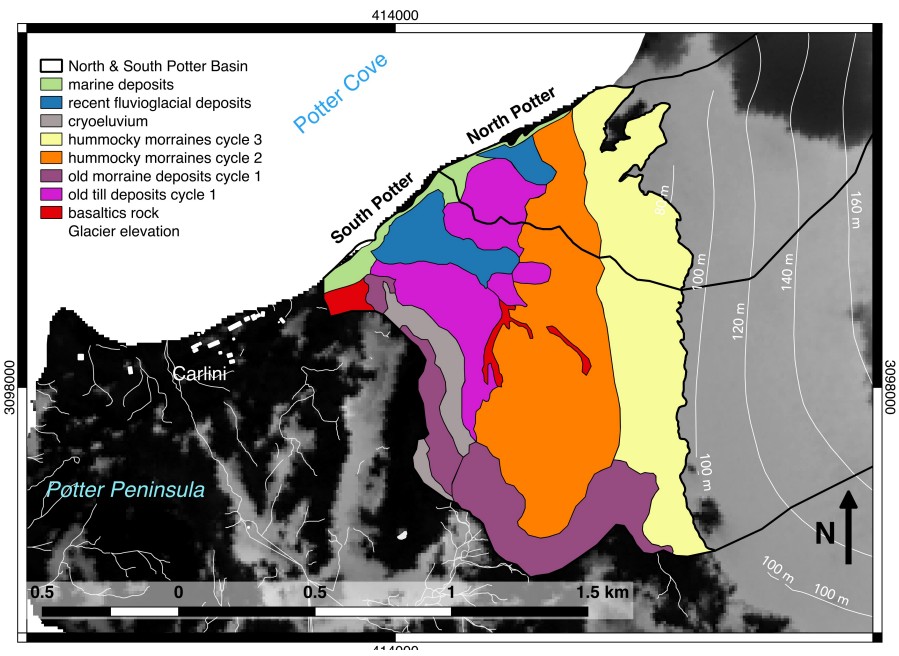

**Figure 2.** Map of distribution of geological deposits from different cycles (1: oldest to 3: most recent) in the Potter Basin, King George Island. Background: SPOT-4, 18 November 2010, ©ESA TPM, 2010

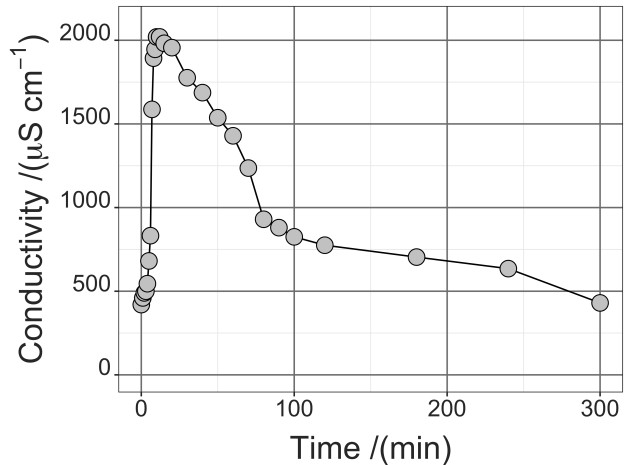

**Figure 3.** Shown is the Darcy permeability test on the basis of Eq. 1 for measurements in the fluvioglacial deposits on Potter Peninsula, King George Island.

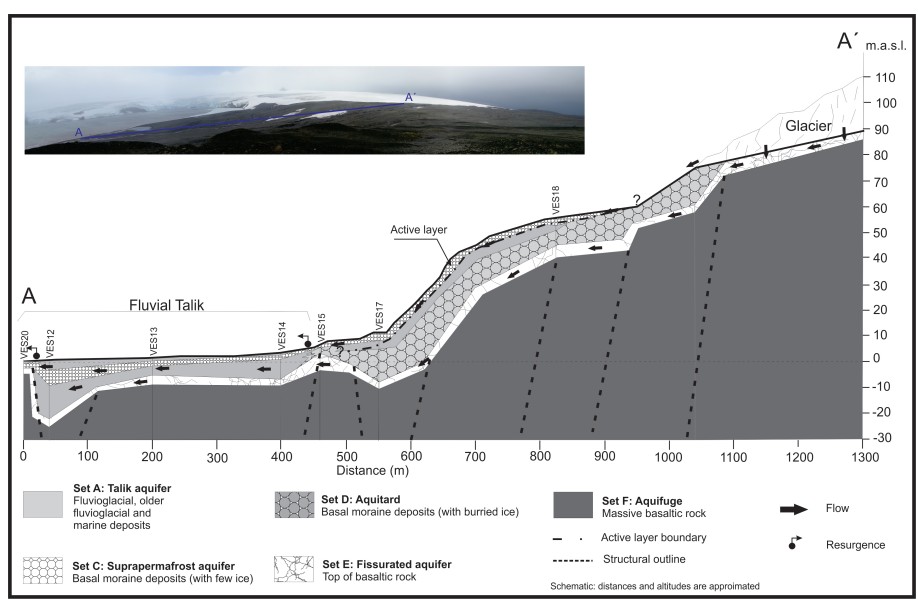

**Figure 4.** Transversal section scheme of the hydrostratigraphy in Potter Basin, King George Island. The photo of Potter Cove in the graph was taken from the Three-Brothers Hill next to Carlini Station.

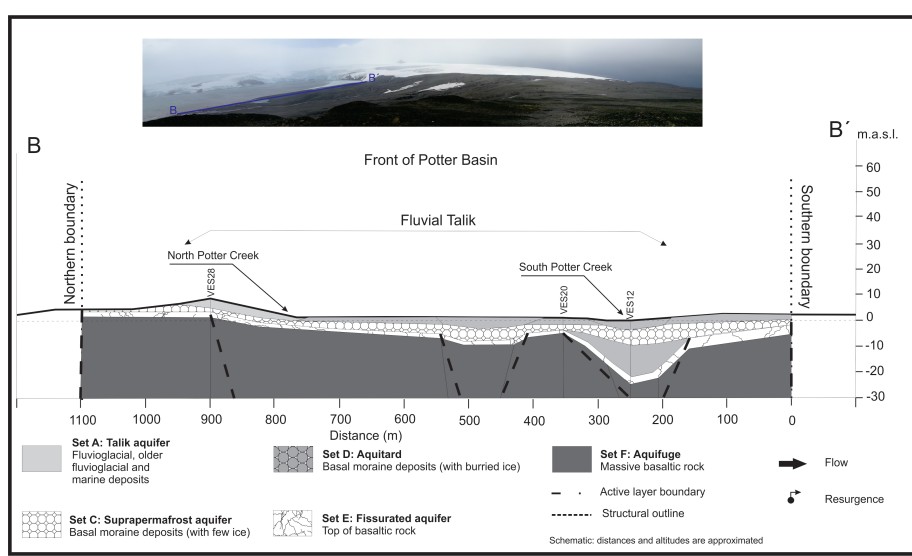

**Figure 5.** Longitudinal section scheme of the hydrostratigraphy in Potter Basin, King George Island. The photo of Potter Cove in the graph was taken from the Three-Brothers Hill next to Carlini Station.

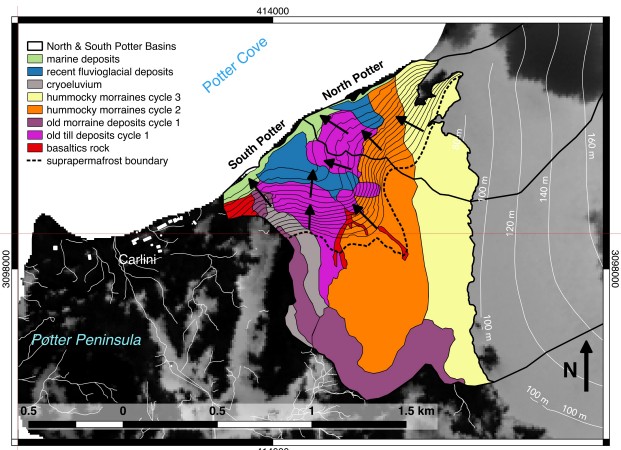

**Figure 6.** Hydrogeological map of suprapermafrost area and fluxes in the Potter Basins, King George Island. Background: SPOT-4, 18 November 2010, ©ESA TPM, 2010

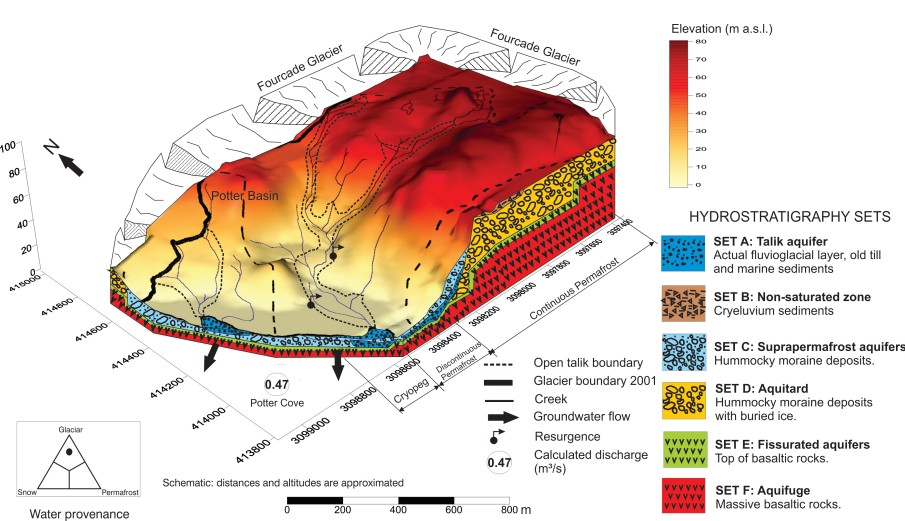

**Figure 7.** Hydrostratigraphic 3D-schematic of Potter Basin, King George Island.

**Table 1.** Analysis of geoelectrical observations, relation between resistive layers and geological interpretation of VES measurements (January 2011 to March 2011) in the Potter basin, King George Island, AP.

| Layer | Thickness /m | Resistivity /(W m) | Proposed Correlation |
|-------|--------------|--------------------|-----------------------|
| I | 1.18 | 76.8 | Coastal deposits and gravel beach ridge |
| II | 1.27 | 901.0 | Crioeluvium |
| III | 1.74 | 943.8 | Acutal fluvioglacial deposits |
| IV | 2.0 | 1308.9 | Hummocky moraines deposits (little or discontinuous ice) |
| V | 5.21 | 1316.7 | Old till deposits (hummocky moraines, fluvioglacial) |
| VI | 4.28 | 5180.8 | Hummocky moraine deposits (with buried ice) |
| VII | 2.89 | 8005.4 | Basaltic rock (fissured and weather alteration) |
| VIII | – | 26761.6 | Basaltic rock (few fissures) |

**Table 2.** Hydrostratigraphic interpretation of observations in the Potter basin, King George Island, AP.

| Group | Layer | Hydrostratigraphy | Permeability |
|-------|-------|-------------------|--------------|
| A | II + V | Talik and cryopeg (fluvioglacial deposits) | 5.231 |
| A | I | Talik and cryopeg (marine deposits) | 1.389 |
| B | III | Unsaturated active layer (only snow infiltration) | no data |
| C | IV | Suprapermafrost aquifer (glacial water supply) | 0.371 |
| D | VI | Aquitard (moraine rich with buried ice) | no data |
| E | VII | Fissurated aquifer (top basalt) | 0.0131 |
| F | VIII | Aquifuge (massive basalt) | no data |

**Table 3.** Specific yield or porosity analysis for geologic units in the Potter basin, King George Island, AP.

| Geologic units | Specific yield dimensionless |
|----------------|------------------------------|
| Coastal depeosits and gravel beach | 0.31 |
| Cryoeluvium | > 0.4 |
| Actual fluvioglacial deposits | 0.12 |
| Hummocky moraines deposits | 0.06 |
| Old till deposits (hummocky moraines, fluvioglacial | 0.12 |
| Basaltic rock (fissured and weather alteration) | 0.01 |

**Table 4.** Hydraulic parameters and gradients calculated for the talik and suprapermafrost aquifers in the Potter basin, King George Island, AP.

| Group | Lithological type | Aquifer surface area /(m$^2$) | Thickness /m | Permeability /(m s$^{-1}$) | Transmissivity /($10^{-3}$ m$^2$ s$^{-1}$) | Hydraulic gradient /(m m$^{-1}$) |
|-------|-------------------|-------------------------------|--------------|---------------------------|---------------------------------------------|----------------------------------|
| A | Actual fluvioglaciar | 544.18 | 5.231 | 1.74 | 9.097 | 0.066 |
| A | Old till sediments | 533.46 | 5.231 | 5.2 | 27.199 | 0.079 |
| A | Marine sediments | 280.15 | 1.389 | 1.18 | 1.620 | 0.029 |
| C | Moraine | 1411.73 | 0.372 | 2 | 0.752 | 0.093 |
| E | Fissurated | 995.35 | $1.308 \cdot 10^{-2}$ | 3 | 0.046 | 0.079 |

**Table 5.** Characteristics of the active layer and maximum summer normative thawing depth for the Potter basin, King George Island, AP, according to the empirical relationship for normative paratmeters of seasonal defrosting by Khrustalev(2005) in Eq. 3 to 7.

| Month | $t_p$ /(°C) | $t_1$ /(°C) | $\tau_1$ /hour | $q$ | $k_{soil}$ /(kcal °C m$^{-2}$ hour$^{-1}$) | $H_d$ /m |
|-------|------|------|--------|---------|---------|------|
| December | 1.54 | 4.556 | 1037.4 | 870.905 | -158.556 | 4.29 |
| January | 1.59 | 4.626 | 966.05 | 844.455 | -74.687 | 4.24 |
| February | 0.61 | 3.254 | 843.0 | 787.337 | 42.730 | 3.44 |
| March | 0.88 | 3.632 | 594.60 | 710.169 | 213.945 | 3.21 |

**Table 6.** Calculated aquifer discharge from Potter basin to Potter Cove, King George Island, AP.

| Group | Lithology | Permeability /($10^{-3}$ m s$^{-1}$) | Aquifer section /(m$^2$) | $\nabla_i$ /(m m$^{-1}$) | $V_d$ /($10^{-6}$ m s$^{-1}$) | $Q$ /($10^{-3}$ m$^3$ s$^{-1}$) |
|-------|-----------|--------------------------------------|--------------------------|--------------------------|-------------------------------|---------------------------------|
| A | Actual fluvioglacial deposit | 5.231 | 544.18 | 0.066 | 345.25 | 187.89 |
| A | Old till deposit | 5.231 | 533.46 | 0.079 | 413.19 | 220.47 |
| A | Marine sediment | 1.389 | 280.15 | 0.029 | 40.28 | 11.28 |
| C | Moraine | 0.372 | 1411.73 | 0.093 | 32.64 | 48.78 |
| E | Fissurated aquifer | 0.013 | 995.35 | 0.079 | 1.03 | 1.04 |

**Table 7.** Calculated aquifer discharge from Potter basin to Potter Cove, King George Island, AP.

| Group | Lithologic type | $A$ /($10^3$ m$^2$) | $b_{sat}$ /m | $\phi_e$ /(m$^3$ m$^{-3}$) | $R_t$ /($10^3$ m$^3$) |
|-------|-----------------|---------------------|--------------|----------------------------|------------------------|
| A | Actual fluvioglacial aquifer | 319.879 | 1.39 | 0.12 | 53.3 |
| A | Old till deposit | 491.878 | 4.85 | 0.12 | 286.2 |
| A | Marine sediment | 660.399 | 1.65 | 0.06 | 65.3 |
| C | Moraine | 5724.108 | 2.65 | 0.06 | 151.6 |
| E | Fissurated aquifer | 141.308 | 0.83 | 0.01 | 11.7 |