# Peer review of "Discharge of groundwater flow to the Potter Cove on King George Island, Antarctic Peninsula"

_Hydrology and Earth System Sciences, 2020_

## Referee Comment (RC1) · Anonymous Referee #1 · 9 Nov 2020

General Comments

The paper analyzed is very interesting, aims to establish a conceptual hydrogeological model for then glacier ablation and groundwater discharge in the northern antarctic peninsula region. The studio is focused on the Potter Cove on King George Island. Most of the work dealing with this issue is addressed from a very large scale, however, this work is done from the scale of a small watershed. The work simultaneously applies numerous hydrogeological tools including: in situ observations, remote sensing, geologic and geomorphologic approach, aerial images, GPS data, Vertical electrical soundings etc.

The objectives are well-thought out and clear. The conclusions are also clear, useful, and well-exposed. From a methodological point of view it is a correct job. It is very interesting how the problem has been addressed, using various complementary techniques.

In my opinion the main weaknesses are: a) it is an overly local work, b) as the authors say it is only representative of a very short period of time, approximately month and a half, c) is a very speculative work, the experimental part and direct measures are scarce. d) A lot of information about the analytical techniques used is lacking and analytical data for most of the variables used are not provided. (e) the bibliography review is poor.

However, I believe that it is a quality work, publishable and that it can be easily improved to achieve the quality required for its publication.

Specific Comments

[Page 3, Study área] The geological description of the study area is very interesting, although there is a lot of superfluous information from the point of view of the objectives and methodology of the paper, for example the ages of the geological strata.

[Page 4 – Line 8] The text says: "The permafrost found here is comparatively warm (mean annual ground temperatures are greater than -2.0". If compared to permafrost elsewhere, it should be illustrated with some numerical data or some bibliographic reference.

[Page 4 – Line 13] This line is set as an example. The paper frequently cites works by Ermolin and Silva Busso (the text contains 18 self-references) to support the statements described by other authors much earlier. A thorough review of the scientific literature written in English on glaciology should be done.

[Page 5 – Line 5] The aims of the paper are described in the "Data and methods" section. I believe that a better place to present the objectives of the paper is in a

specific paragraph or at the end of the introduction.

[Page 5 – Line 19] The VES technique has been used to determine the vertical structure of the aquifer. How has it been shown that the interpretation given to the values of resistivity corresponds to reality? The mechanical perforation used to perform these interpretations should be displayed. If no mechanical drilling have been performed, the way in which the correlation has been made should be indicated in more detail.

[Page 5 – Line 32] How deep are those perforations? What materials have been found in these perforations? Additional information on such perforations should be provided.

[Page 6 – Line 17] The following sentence is not understood: "The groundwater hydraulic gradient was calculated on the basis of the different hydrogeologic units obtained from the piezometric map." Have the hydrogeological units been obtained from the piezometric map? This should be better explained.

[Page 6 – Line 18, 19] Page 6 reads: "The meteorological, permafrost and glaciological data sets were used for ...." Where can the reader find those Data Sets?. Those Data Sets must be available for study.

[Page 6 – Line 22,23] The statement: "These assumptions are valid during 1 to 1.5 months in the austral summer (presumably January and February) " should be better explained, and if possible, it would be very interesting to know if global warming will cause that period to be extended.

[Page 7 Results, Geological deposits] I do not think this section corresponds to the results section. I'd be better off in the introduction or in section 3. It does not seem like a result, it seems an explanation based on previous work and in the bibliography.

[Page 8 – Line 2, 3] Where has that porosity value been obtained? How has this porosity value been obtained?. How many samples have been analyzed?

[Page 8 – Lines 16, 17] This statement: "These resistive layers can be interpreted as old till deposits of more ancient hummocky moraines or previous fluvioglacial events"

and similar ones should be justified by some kind of supplementary data. For example, line 20 says "it contains marine deposits" Is there any evidence or is it just a guess?

[Page 8 – Line 33] ".....  the determination of in-situ permeability of each group with aquifer .....". The permeability determination has been made on the outcropping material. Can it be said that in depth it will have the same permeability?, the outcropping material will be altered and will be more permeable than the same material in depth. This should be explained better.

[Page 9 – Lines 17] "The results are presented in Table 3" it's already said on line 4

[Page 9 – Lines 17, 18] " ….  and the obtained values are within the typical ranges for such types of lithologies …." Where are these types of litologies said to show this range of values? This must be better justified and quoted in the literature where it can be consulted.

[Page 9 – Lines 33, 34] How has the topographic gradient subglacial been estimated? Why has it been estimated and not measured?

[Page 10 – Lies 1, 4] "On the other hand, there has been a high similarity between piezometric and topographic gradients in the Potter Basin and adjacent Matías Basin on the Potter Peninsula (Silva-Busso, 2009). Based on the above argumentation, topographical gradients instead of the hydraulic gradients are used here as input." This seems like a circular argument. To know that there is a strong correlation between topographic gradient and piezometric gradient you have to know both. If both are known it is no longer necessary to rely on correlation. Can data simply be extrapolated from one basin to another? This should be explained better.

[Page 10 – Lines 29, 30] "The application of the method after Khrustalev (2005) requires a percentage of positive degree days per month higher than that value." If this is the case then the proposed method is not applicable. This statement should be better explained.

[Figure]

[Page 10 – Lines 30,31] " …. taken from Silva Busso and Yermolin (2014)." This publication does not list how these values have been measured. It should be better justified from where these values have been obtained and whether they are estimated or measured.

[Page 11 – Lines 12, 13] Are the results presented in this paper only applicable to the month of February?. There should be detailed weather information from the Potter Bay area. If possible, the dependence between flow and temperature should be better explained. How will global warming affect the system? Can it be quantified how global warming will affect flows?

[Page 11 – Line 14] the next sentence should be better explained: "Little can be inferred about the hydraulic type."

[Equation 8] Where have the sand, silt and clay values been obtained? How many samples have been analyzed?. Analytical data should be available.

[Page 11 – Line 20] How has porosity been measured? With what uncertainty has porosity been measured? . Analytical data should be available.

[Page 13 – Line 30, 31] If litologies are inconsistent, how is it justified to use the same empirical relationships proposed by Bourbié?

[Page 13 – Line 2, 3] The "sensitivity analysis" section should answer the following questions: What is the reason for doing a sensitivity analysis? What its usefulness? What have the sensitivity analysis results been used for? What does the sensitivity analysis have to do with the hydrogeological scheme proposed in Figure 7? Why is the hydrogeological scheme included in the sensitivity analysis section (page 14, line 1)?

[Section 5 Discussion and Conclusions] The values of transmissivity, water velocity, water discharge, should be given as a range not as an exact value. Conclusions on changing seawater quality and its potential impact should be better argued and supported by bibliographic data. Is 0.43 m3 s-1 really significant as freshwater discharge

into the sea? Is more fresh water spilled to the sea now than before warming? What are those biological changes? Is there any evidence of biological changes?

[Table 1] Based on which data the correlations have been made. Are there mechanical drillings to validate them?. How have they been validated?

[Table 2] Permeability data is provided with three decimal places, this is very optimistic. The range in which this value is moved should be provided. It is necessary to adjust the number of decimal places to the precision and error of the technique used.

[Table 3] How many tests have this data been obtained with? What is the uncertainty of the tests? For example, has thickness been accurately measured in millimeters?

[Figure 1] VESs are not installed.

[Figure 6] Is there enough continuity in the suprapermafrost aquifer to be able to draw the isolines as they have been drawn? This point should be better explained.

Technical Corrections [Page 3 – Line 15] piezometric sonde – water level meter (contact gauge?)

[Page 5 Line 2] Potter

---

## Referee Comment (RC2) · Anonymous Referee #2 · 25 Mar 2021

Hydrological research for Antarctic Peninsula (AP) is very important due its vulnerability to climate change. However, related studies are relatively few due to limited data records for the Antarctica. This study calculated the glacial runoff and groundwater flow discharge on a small hydrological catchment Potter Basin in the edge of AP. The value of this study lies on improving our knowledge for the hydrological cycle in the edge of AP and potentially revealing the biogeochemical effects associated with the water cycle over this region. I have some suggestions that may improve the quality of this article.

Major comments:

1. The authors stressed the importance of climate change on the hydrological cycle for the study domain. However, they did not actually discuss the impacts based on their computed data. This much reduced the meaningfulness and value of the article. Though I understand it is difficult to collect related data to plot a time series to show the change of runoff during climate change, I recommended the authors to explore more about it using both their data and previous published research.

2. This is a very local work. To add value on this study, I recommend the authors to extrapolate the knowledge we got from this local site to the whole AP or even the cryosphere of Earth. I think this is also required by HESS journal to show a universal value that can benefit more generic readers.

Specific comments:

P1, L10, "...2719.9 10-5...": Please use symbol "×" at here, also for the other places across the manuscript.

P6, L17-18, "The groundwater hydraulic gradient...obtained from the piezometric map.": Please show more detail for how to get the groundwater hydraulic gradient.

P10, L3-4, "Based on the above...used here as input.": Please discuss uncertainties caused by using topographical gradients instead of hydraulic gradients in the model computation.

P10, L7, "criopeg": cryopeg?

P13, L7-9, for equations (13) and (14): It is not clear that how the authors got the $Q_{max}$, $Q_{min}$, $R_{t,max}$ and $R_{t,min}$, and how they transferred the range of $Q$ and $R$ to the uncertainty range of parameters.

---

## Author Comment (AC1) · 29 Apr 2021

Response to major (structural) points:

We thank the reviewer for the thorough evaluation of our manuscript and the many, very helpful comments to it. The paper includes various complementary techniques to draw conclusions on the hydrological flow regime. We incorporated the comments in the text to emphasise the representativity of this local study to the wider region of the Antarctic Peninsula, and put it into clear context to global climate change. Thanks again for the comprehensive reading and detailed advice.

We included a paragraph into the discussion to put our local study into the context of climate change in the region of the Antarctic Peninsula. This should explain the importance and representativity of this study for the wider region. Field observations in Antarctica are not easily conducted due to the limited accessibility and work under extreme conditions. Compared to many studies in this region, we combine a several fieldwork-intensive measurements in this study. Melt periods during austral winter months are becoming more frequent due to the ongoing change process. We included a paragraph into the discussion section to elaborate the applicability of the study results to other time periods outside the short time period of austral summer. This manuscript is the synthesis of field measurements over two months on King George Island and of former papers where extensive data have been analysed and discussed, that are cited in the text and are freely available (see section "data availability"). We are thankful for the notification on missing data, and included them into the manuscript (see specific comments) when not available online (see "data availability" and above). We included bibliography review on glaciology, climatology and where necessary as reference to the implications of ocean currents and marine biota.

Response to specific points:

[Page 3, Study área] The geological description of the study area is very interesting, although there is a lot of superfluous information from the point of view of the objectives and methodology of the paper, for example the ages of the geological strata.

Geological information is the basis of geoelectric interpretation. Reducing the information too much can result in failure to understand the interpretation. We eliminated information where it (in our view) was not partial to the understanding.

[Page 4 – Line 8] The text says: "The permafrost found here is comparatively warm (mean annual ground temperatures are greater than -2.0". If compared to permafrost elsewhere, it should be illustrated with some numerical data or some bibliographic reference.

We included the necessary bibliographic references and put our statement into global context. Thanks for the advice!

[Page 4 – Line 13] This line is set as an example. The paper frequently cites works by Ermolin and Silva Busso (the text contains 18 self-references) to support the statements described by other authors much earlier. A thorough review of the scientific literature written in English on glaciology should be done.

We included a thorough review on glaciology (and climate change) of the wider region. The Potter Peninsula is not frequented by a lot of other research groups, so the amount of different authors for the study site is limited.

[Page 5 – Line 5] The aims of the paper are described in the "Data and methods" section. I believe that a better place to present the objectives of the paper is in a specific paragraph or at the end of the introduction.

You are right, and we moved the paragraph.

[Page 5 – Line 19] The VES technique has been used to determine the vertical structure of the aquifer. How has it been shown that the interpretation given to the values of resistivity corresponds to reality? The mechanical perforation used to perform these interpretations should be displayed. If no mechanical drilling have been performed, the way in which the correlation has been made should be indicated in more detail.

Resistivity is an indirect measure of changes in water, ice and rock. There are no parametric wells, so the interpretation was correlated with the ground cuts in the stream crankcases and wells that were performed for the recognition of the suprapermafrost aquifer. Deeper interpretation are drawn from observations of the geological characteristics and at the edge of the basin.

[Page 5 – Line 32] How deep are those perforations? What materials have been found in these perforations? Additional information on such perforations should be provided.

They are a total of 10 holes between 1.25-2 meters deep, and are, generally, divided

into two groups: coastal (below 3 m a.s.l.) and higher wells (above). We included a summary of the detailed geological description.

[Page 6 – Line 17] The following sentence is not understood: "The groundwater hydraulic gradient was calculated on the basis of the different hydrogeologic units obtained from the piezometric map." Have the hydrogeological units been obtained from the piezometric map? This should be better explained.

Thanks for the comment, we elaborated and changed the phrasing.

[Page 6 – Line 18, 19] Page 6 reads: "The meteorological, permafrost and glaciological data sets were used for ...." Where can the reader find those Data Sets?. Those Data Sets must be available for study.

They are referred to in the section "data availability"

[Page 6 – Line 22,23] The statement: "These assumptions are valid during 1 to 1.5 months in the austral summer (presumably January and February) " should be better explained, and if possible, it would be very interesting to know if global warming will cause that period to be extended.

There are preceding studies that show the importance of winter melt periods, where the conditions for applicability are met. Global warming does not cause a general extension of austral summer, but due to climate circulations, the region gets more often under the influence of mid-latitude weather patterns during winter time, where conditions for applicability are met more often. We included a paragraph in the discussion.

[Page 7 Results, Geological deposits] I do not think this section corresponds to the results section. I'd be better off in the introduction or in section 3. It does not seem like a result, it seems an explanation based on previous work and in the bibliography.

You are right, and we moved the paragraph.

[Page 8 – Line 2, 3] Where has that porosity value been obtained? How has this

porosity value been obtained?. How many samples have been analyzed?

With a Slug Test and Le Franc method (point permeability test) and with the method of using NaCl as a plotter in two wells one load and one reading salinity. We elaborated and rephrased the respective parts on porosity measurements.

[Page 8 – Lines 16, 17] This statement: "These resistive layers can be interpreted as old till deposits of more ancient hummocky moraines or previous fluvioglacial events" and similar ones should be justified by some kind of supplementary data. For example, line 20 says "it contains marine deposits" Is there any evidence or is it just a guess?

The ancient glaciofluvial plain do not have outcroppings, so it can't be sure and that's why it's an interpretation of resistivity from the basaltic bed rock that emerges at the edges of the basin. It is an educated guess based on recent glaciofluvial plains where they emerge in the basin, and where measurements can be carried out. Previous studies show that in the preceding geological period the position of the glacier front on land was different and closer to the sea. Similar but slightly older deposits may be in depth buried by the more modern ones. Due to the recent local geology that covers it all this is an acceptable possible interpretation.

[Page 8 – Line 33] "..... the determination of in-situ permeability of each group with aquifer .....". The permeability determination has been made on the outcropping material. Can it be said that in depth it will have the same permeability?, the outcropping material will be altered and will be more permeable than the same material in depth. This should be explained better.

We rephrased and elaborated in more detail.

[Page 9 – Lines 17] "The results are presented in Table 3" it's already said on line 4

This was a problem with the cross-referencing in Latex, solved by rerunning the document processing.

[Page 9 – Lines 17, 18] " . . .. and the obtained values are within the typical ranges

for such types of lithologies . . .." Where are these types of litologies said to show this range of values? This must be better justified and quoted in the literature where it can be consulted.

We rephrased and put in the necessary citation. Thanks for the comment!

[Page 9 – Lines 33, 34] How has the topographic gradient subglacial been estimated? Why has it been estimated and not measured?

We recognize that estimating the topographic gradient of the subglacial bedrock is not exact, but there exists no data on bedrock topography up to today. The only way to get more precise data on this parameter is to engage in a campaign on measuring glacier bedrock topography with ground-penetrating radar. The best option to get an estimate of this parameter is to our knowledge, to obtain the parameters of topography for the bedrock of the ice-free area, and by geolectrics the relief of the roof of the bed rock. We assume, that being an average value and considering that bed rock does not change its depth too much under the glacier is a first approximation of the actual gradient.

[Page 10 – Lies 1, 4] "On the other hand, there has been a high similarity between piezometric and topographic gradients in the Potter Basin and adjacent Matías Basin on the Potter Peninsula (Silva-Busso, 2009). Based on the above argumentation, to-pographical gradients instead of the hydraulic gradients are used here as input." This seems like a circular argument. To know that there is a strong correlation between topographic gradient and piezometric gradient you have to know both. If both are known it is no longer necessary to rely on correlation. Can data simply be extrapolated from one basin to another? This should be explained better.

This is a misunderstanding, probably due to poor phrasing, since both values were measured or approximated by measurements. We rephrased the paragraph to make it clearer.

[Page 10 – Lines 29, 30] "The application of the method after Khrustalev (2005) requires a percentage of positive degree days per month higher than that value." If this is the case then the proposed method is not applicable. This statement should be better explained.

We rephrased this paragraph to elaborate on the applicability.

[Page 10 – Lines 30,31] " ....  taken from Silva Busso and Yermolin (2014)."  This publication does not list how these values have been measured.  It should be better justified from where these values have been obtained and whether they are estimated or measured.

We included the necessary citation and where values were taken from.

[Page 11 – Lines 12, 13] Are the results presented in this paper only applicable to the month of February?.  There should be detailed weather information from the Potter Bay area. If possible, the dependence between flow and temperature should be better explained. How will global warming affect the system? Can it be quantified how global warming will affect flows?

Thanks for making us aware of this point. We included a paragraph into the discussion to elaborate on applicability of this study to other time periods, and also to put our study into the context of global warming.

[Page 11 – Line 14] the next sentence should be better explained: "Little can be inferred about the hydraulic type."

We assume that it is a free aquifer for its hydrogeological and geocryological characteristics, but the hydraulic type can only be demonstrated with a pumping test, which was not possible here. However, the probabilities that the aquifer be free outweigh the other possibilities (semi-confined or confined).  In addition, they have been verified to be free in other places such as the neighboring Matías Basin in Silva Busso (2009). We rephrased the paragraph on hydraulic type.

[Equation 8] Where have the sand, silt and clay values been obtained?  How many

samples have been analyzed?. Analytical data should be available.

Thanks for making us aware of this issue! The requested information was included into the text.

[Page 11 – Line 20] How has porosity been measured? With what uncertainty has porosity been measured? . Analytical data should be available.

The measurement principal was included into the text, and the laboratory results into a Table.

[Page 13 – Line 30, 31] If litologies are inconsistent, how is it justified to use the same empirical relationships proposed by Bourbié?

The difficulty arises because of the contact of the bed rock with the basaltic rock, that can be altered and possess a certain degree of porosity. The magnitude would be different from the cracked part of the rock or the massive rock. This discontinuity is typical for basaltic rocks. We believe that the best reference is that of Bourbie et.al., 1987 despite existing inconsistencies. We rephrased to make our point more clear.

[Page 13 – Line 2, 3] The "sensitivity analysis" section should answer the following questions: What is the reason for doing a sensitivity analysis? What its usefulness? What have the sensitivity analysis results been used for? What does the sensitivity analysis have to do with the hydrogeological scheme proposed in Figure 7? Why is the hydrogeological scheme included in the sensitivity analysis section (page 14, line 1)?

The main objective of this analysis is to verify how important and significant the result for basaltic rock is compared to other flows. It is used to verify the variability of the flow in basaltic rock and whether to take it into account for the calculation of the total flow. The sensitivity analysis is used to calculate the total flow of groundwater discharge into Potter Cove. The reference for Figure 7 was included into the discussion section.

[Section 5 Discussion and Conclusions] The values of transmissivity, water velocity, water discharge, should be given as a range not as an exact value. Conclusions on

changing seawater quality and its potential impact should be better argued and supported by bibliographic data. Is 0.43 m3 s-1 really significant as freshwater discharge into the sea? Is more fresh water spilled to the sea now than before warming? What are those biological changes? Is there any evidence of biological changes?

The values are expressed as ranges and differentiated by aquifer type in the respective data tables. All measurements and data values were converted to SI units, thereby it appears to be overly precise, but it is within the precision of the observational techniques.

[Table 1] Based on which data the correlations have been made. Are there mechanical drillings to validate them?. How have they been validated?

This also refers to a former comment, and we included a more detailed description of technique and observation. There are also the outcrops of the different deposits in the stream valley. The two deepest units are basalt bedrock, which emerges on the banks of the basin, and the ancient fluvioglacial plain. The latter is interpreted as a logical consequence of the geological evolution of the environment. For this reason we use the indirect geoelectric observation technique like geoelectric.

[Table 2] Permeability data is provided with three decimal places, this is very optimistic. The range in which this value is moved should be provided. It is necessary to adjust the number of decimal places to the precision and error of the technique used.

The seemingly high precision of permeability data arise from the conversion to SI units. The least significant bit converts then into three decimal places and reflect the precision of the observations.

[Table 3] How many tests have this data been obtained with? What is the uncertainty of the tests? For example, has thickness been accurately measured in millimeters?

A proper uncertainty assessment is difficult to address in hydrogeological studies, that usually rely on scarce data and, in parts, on interpretation. We are not aware of any

hydrogeology paper up to date, that includes this, especially not in studies in remote areas with very low accessibility.

[Figure 1] VESs are not installed.

VES's are installed and marked by yellow circles.

[Figure 6] Is there enough continuity in the suprapermafrost aquifer to be able to draw the isolines as they have been drawn? This point should be better explained.

This was previously explained in the paper. It was said that up to 45 m a.s.l. could be considered relatively continuous so it is the last marked isofoot and it is in dotted lines which for any geologist clearly indicates that it is a data that can present modifications. Above 45 masl we cannot be sure of the continuity of the aquifer and the permafrost is more continuous

Technical Corrections

[Page 3 – Line 15] piezometric sonde – water level meter (contact gauge?)

Piezometric probe.

Please also note the supplement to this comment:
https://hess.copernicus.org/preprints/hess-2020-422/hess-2020-422-AC1-supplement.pdf

---

## Author Comment (AC2) · 29 Apr 2021

Response to major (structural) points:

We thank the reviewer for the thorough evaluation of our manuscript and the helpful comments to it. The paper includes various complementary techniques to draw conclusions on the hydrological flow regime. We incorporated the comments in the text to emphasise the representativity of this local study to the wider region of the Antarctic Peninsula, and put it into clear context to global climate change. Thanks again for the comprehensive reading and advice.

Thank you very much for this major comment. We elaborated the context of our study results with respect to climate change. In especially, the more frequent melt periods observed by climatological and glaciological studies, indicate the basis for change processes of e.g. marine biota (highlighted by Braeckman et al, 2021) or ocean currents (e.g. Meredith et al. 2018). We included and elaborated at several points in the manuscript, following the very valuable and helpful comments of both referees.

Hydrogeological field studies are mostly local studies, in especially in this remote extreme area with very limited accessibility. We elaborated mainly in the discussion section the representativity of the local study to the wider region and its implications. We hope, this puts our main conclusions into a more general and to shows as well that, although a local study during a limited time period, the results are applicable to the wider Peninsula region and that due to ongoing climatic change, this study serves as a model for future scenarios of hydrological catchments along the rugged western coast of the Antarctic Peninsula.

Response to specific points:

P1, L10, ". . .2719.9 10-5. . .": Please use symbol "$\times$" at here, also for the other places across the manuscript.

We agree. Thanks a lot for this comment.

P6, L17-18, "The groundwater hydraulic gradient...obtained from the piezometric map.": Please show more detail for how to get the groundwater hydraulic gradient.

The simplest way is to make the hydraulic load difference between two isocurves divided by the linear distance separating them. It should be ensured that the measurement is perpendicular to both curves. We use this shape because the basin is small. We rephrased to make this point clear.

P10, L3-4, "Based on the above. . .used here as input.": Please discuss uncertainties caused by using topographical gradients instead of hydraulic gradients in the model
computation.

A proper uncertainty assessment is difficult to address in hydrogeological studies, that usually rely on scarce data and, in parts, on interpretation. We are not aware of any hydrogeology paper up to date, that includes this, especially not in studies in remote areas with very low accessibility. We included a paragraph on estimation of topographical and hydraulic gradients. The sensitivity analysis is meant to address the variability of results when varying the input. We changed this paragraph according to this comment and comments of Referee #1.

P10, L7, "criopeg": cryopeg?

Cryopeg. Thanks! The manuscript was first written in Spanish, and there were few artefacts like this.

P13, L7-9, for equations (13) and (14): It is not clear that how the authors got the Qmax, Qmin, Rt,max and Rt,min, and how they transferred the range of Q and R to the uncertainty range of parameters.

The min-max values arise from several measurements at different locations, and reflect the variability of the observed quantity. They then translate into a min-max range of derived variables. We elaborated this in the text to make our approach more transparent.

Please also note the supplement to this comment:
https://hess.copernicus.org/preprints/hess-2020-422/hess-2020-422-AC2-supplement.pdf